# Vitamin C Deficiency and the Risk of Osteoporosis in Patients with an Inflammatory Bowel Disease

**DOI:** 10.3390/nu12082263

**Published:** 2020-07-29

**Authors:** Alicja Ewa Ratajczak, Aleksandra Szymczak-Tomczak, Marzena Skrzypczak-Zielińska, Anna Maria Rychter, Agnieszka Zawada, Agnieszka Dobrowolska, Iwona Krela-Kaźmierczak

**Affiliations:** 1Department of Gastroenterology, Dietetics and Internal Diseases, Poznan University of Medical Sciences, 60-355 Poznan, Poland; aleksandra.szymczak@o2.pl (A.S.-T.); a.m.rychter@gmail.com (A.M.R.); a.zawada@ump.edu.pl (A.Z.); agdob@ump.edu.pl (A.D.); 2Institute of Human Genetics, Polish Academy of Sciences, 60-479 Poznan, Poland; mskrzypczakzielinska@gmail.com

**Keywords:** vitamin C, inflammatory bowel disease, osteoporosis, supplementation, gastrointestinal microbiota, diet, glutathione transferase

## Abstract

Recent research studies have shown that vitamin C (ascorbic acid) may affect bone mineral density and that a deficiency of ascorbic acid leads to the development of osteoporosis. Patients suffering from an inflammatory bowel disease are at a risk of low bone mineral density. It is vital to notice that patients with Crohn’s disease and ulcerative colitis also are at risk of vitamin C deficiency which is due to factors such as reduced consumption of fresh vegetables and fruits, i.e., the main sources of ascorbic acid. Additionally, some patients follow diets which may provide an insufficient amount of vitamin C. Moreover, serum vitamin C level also is dependent on genetic factors, such as *SLC23A1* and *SLC23A2* genes, encoding sodium-dependent vitamin C transporters and *GSTM1*, *GSTP1* and *GSTT1* genes which encode glutathione S-transferases. Furthermore, ascorbic acid may modify the composition of gut microbiota which plays a role in the pathogenesis of an inflammatory bowel disease.

## 1. Introduction

Vitamin C (ascorbic acid, AA) is an antioxidant, a water-soluble vitamin produced by plants and most of animals. The human body is not capable of producing vitamin C which stems from the lack of a functional gene coding i-gulonolactone oxidase, i.e., an essential enzyme for the biosynthesis of ascorbic acid [1,2]. Ascorbic acid acts as an electron donor for both enzyme and non-enzyme reactions. Additionally, vitamin C is essential for the synthesis of collagen and carnitine. Furthermore, AA protects against mutation and has a protective effect on white blood cells [3]. Ascorbic acid participates in carbohydrates metabolism, melanin synthesis and recreation of vitamin E and glutathione [4,5]. The main sources of vitamin C are vegetables and fruits, particularly citrus, kiwifruit, mango, broccoli, tomatoes, strawberries, parsley, potatoes and pepper. It is worth stressing that both the storage and processing of food decreases vitamin C content. Recommended dietary allowances for vitamin C (for adults; age 19 years and more) is 65 mg and 75 mg for women and men, respectively. This is turn suggests that the consumption of 5–9 portions of fresh and low-processed vegetables and fruits provides about 200 mg of vitamin C [2,6,7,8]. AA is absorbed in the small intestine and about 70–90% of vitamin C is absorbed in the daily supply of 200 mg. However, a higher dose leads to a decreased absorption and only 33% of vitamin C is absorbed when the dose is increased to 1250 mg [7]. Absorption of vitamin C takes place through facilitated diffusion (using GLUT—glucose transporters) and active transport (using SVCT—sodium vitamin C cotransporters) [9]. Moreover, vitamin C deficiency leads to scurvy, which may be manifested as gingival swelling, loss of teeth, ecchymosis, poor wound healing, fatigue, vision problems and neurological disorders. Manifestations of vitamin C deficiency are usually related to changes in the extracellular matrix in bone, blood vessels and tendons which leads to bone fragility and hemorrhage [10,11].

## 2. Vitamin C Deficiency in IBD Patients and the Risk of Osteoporosis

Patients suffering from an inflammatory bowel disease (IBD) are at a higher risk of nutritional deficiencies, including vitamin C deficiency. This is due to the increased demand, malabsorption, active inflammation and reduction in consumption of vegetables and fruits rich in vitamin C [12]. The mean serum level of vitamin C in patients with IBD and healthy individuals are presented in Table 1. Patients with IBD present high tumor necrosis factor α (TNF-α) levels, which also decreases absorption of various nutrients, such as vitamin C. Therefore, patients suffering from IBD have a lower concentration of AA [13].

Children suffering from IBD consumed significantly smaller amounts of vitamin C than the healthy peers [14]. According to Lim et al. the intake of vitamin C is lower among the malnourished IBD patients than among the well-nourished patients [15]. Lower intake of vitamin C may be due to a reduced intake of fresh vegetables and fruits. However, Głąbska et al. reported that the intake of vegetables and fruits, as well as of vitamin C does not significantly differ between men suffering from ulcerative colitis (UC) and the healthy individuals [16]. Table 2 shows the average intake of vitamin C among patients with CD or UC and healthy. More UC patients than CD patients presented inadequate intake of vitamin C [17]. Additionally, vitamin C intake is correlated negatively with the risk of IBD [18]. Moreover, AA decreases oxidative stress which is one of the factors of IBD, and may increase the disease symptoms [19].

It is vital to bear in mind that AA also affects BMD. As an antioxidant, vitamin C decreases the level of reactive oxygen species which increase bone resorption. Moreover, ascorbic acid increases the differentiation of osteoclasts [23]. A meta-analysis shows that a higher vitamin C intake was associated with a higher BMD of the femoral neck and lumbar spine (*p* = 0003 and *p* = 0,001, respectively). Additionally, a higher consumption of vitamin C decreased the risk of osteoporosis by 33% [24]. Kim et al. reported that women suffering from osteoporosis consumed a lower amount of vitamin C than women with normal BMD. However, this association occurred in women aged 50–59 with a deficiency of vitamin D [25]. Thus, a long-term supplementation of vitamin C was linked with a higher BMD. On the other hand, there was no association between a dietary intake of vitamin C and BMD [26]. The association between vitamin C intake and BMD requires further research. It is crucial to notice that some studies concerning the influence of vitamin C on bone mineralization did not into account for other risk factors for osteoporosis.

Recommended dietary allowances for vitamin C (for adults; age 19 years and more) is 65 mg and 75 mg for women and men, respectively, with the Tolerable Upper Intake Levels estimated at 2000 mg. Although, AA has minimal side effects and is well tolerated, high doses may cause gastrointestinal symptoms, such as nausea and diarrhea [6]. According to research by Ferraro et al., a total daily intake was associated with the risk of renal stones in men, but not in women [27]. Following guidelines of team for dietary supplements, 1000 mg/day is the maximum dose of vitamin C in the supplement. Higher doses may lead to various consequences, e.g., nephrolithiasis [28]. Nevertheless, it is vital to notice that the guidelines of the ESPEN (European society for clinical nutrition and metabolism) for patients suffering from IBD do not indicate a need for the supplementation of vitamin C [29].

## 3. Pharmacological Treatment in IBD

A number of medications, such as anti-inflammatory (glucocorticosteroids—GCs, salicylates), immunosuppressive (azathioprine, methotrexate, mercaptopurine) and biologic agents (infliximab, adalimumab, vedolizumab, ustekinumab) are used in the treatment of IBD [30]. Moreover, chronic steroid therapy may decrease BMD, damage bone tissue structure, as well as increase the risk of fractures [31]. In fact, glucocorticosteroids cause imbalance between osteoclasts and osteoblasts activity, inhibit osteoblastogenesis, as well as induce apoptosis of osteoblasts [32]. Moreover, GCs affect bone by means of RANK–RANKL–OPG (receptor activator for nuclear factor κB Ligand–receptor activator for nuclear factor κB-Osteoprotegerin) and increase RANKL expression and decrease expression of OPG receptor in osteoblast. This further leads to a bone resorption and inhibits bone formation [33]. Additionally, GCs affect the indirect gastrointestinal tract, decreasing calcium absorption and inhibit calcium reabsorption in kidneys leading to an increase in the parathormone (PTH) level, responsible for the stimulation of osteoclasts and increasing bone loss [32].

The impact mechanisms of biologic medications on bones are unclear. Anti-TNF (Tumor necrosis factor) may affect bone in bone in different ways [34]. A reduction in TNF decreases the differentiation of osteoclasts’, stimulates their apoptosis and increases osteoblasts differentiation [35]. Moreover, anti-TNF influences on TNF-α, IL-17, IL-6, IL-1 and IL-23, which affect osteoclasts development [36]. According to Ziolkowska et al. anti-TNF therapy decreased OPG and soluble RANKL (sRANKL) in patients with elevated levels of them, but did not change OPG/sRANKL ratio [37]. Anti-TNF therapy may decrease carboxyterminal telopeptide serum level [34]. Additionally, a reduction in the activity of IBD not only leads to an improvement in patients’ physical activity and a stimulation of bone formation, but also to an increase in vitamin D and calcium absorption.

Additionally, infliximab possibly increases the N-terminal telopeptide of type I collagen. However, research regarding the association between the use of infliximab and BMD are contradictory [38].

## 4. Genetic Background for Vitamin C Deficiency

The supply of vitamin C stems from the diet and, to a small extent, can be supported by the production of gut microflora [39]. However, a high vitamin C intake may not increase its intestinal uptake and serum level which can be due to several factors, including diseases, environmental and lifestyle factors, but also genetic polymorphisms. DNA variants affect the homeostasis of vitamin C in two main pathways: (1) altering vitamin C transport (active and facilitated) in the body and (2) changing proteins which suppress oxidative stress or detoxify oxidative damaged biomolecules [40].

### 4.1. Genes Encoded Transport Proteins

The transmembrane active transport mechanisms, which is dependent on sodium ions, consist of two sodium-dependent vitamin C transporters (SVCT): SVCT1 and SVCT2 surface glycoproteins. SVCT1 is primarily limited to vitamin C absorption in the small intestine and to its reabsorption in the kidneys. Vitamin C transport using SVCT2 applies to most body tissues protecting them from oxidative stress. Thus, in the intestine, both SVCT transporters occur, although their distribution is different and they play different roles [41]. SVCT1 is expressed on the apical membrane and it is involved in vitamin C transport across the intestinal barrier [42], whereas SVCT2 is located at the basolateral surface and transports vitamin C from the blood into enterocytes [43]. SVCT1 and SVCT2 are encoded by solute carrier family 23 genes: *SLC23A1* (MIM:603790) and *SLC23A2* (MIM:603791), respectively, which are similar in structure; however, they differ considerably in length. *SLC23A1* gene is located on chromosome 5q31.2–31.3 and consists of 15 exons and span 16 096 bp. On the other hand, *SLC23A2* gene maps to chromosome 20p13-p12 and includes 17 exons and is 158 398 bp long. Both genes are characterized by high genetic polymorphism, although for the most part, their clinical significance has not been demonstrated or has not yet been described. The most important variants related to active transport of vitamin C are diverse and they have been presented in detail by Michels et al. [44]. Variants associated with the deficiency of vitamin C in the human body are summarized in Table 3. Particular attention should be given to the exonic variant rs33972313 (c.790G>A, p.Val264Met) in the *SLC23A1* gene, which was identified as associated with decreased circulating concentrations of ascorbic acid in large studies on the general population (over 15,000 participants) [45]. In the same meta-analysis, the results for rs10063949 (c.-97-487A>G) *locus* have been presented, but the impact of this genetic variant on vitamin C plasma levels have not been confirmed. Interestingly, in the research of 311 patients from the Manitoba IBD Cohort, it has been observed that genotypes AG and GG at rs10063949 increase the risk of Crohn’s disease (CD) (OR: 2.54; 95% CI: 1.38–4.66; OR: 4.72; 95% CI: 2.53–8.81, *p* < 0.001; respectively) [46]. It is worth noting that the functionality and the level of SVCT proteins are not only affected by substitutes for single nucleotides in the coding and intronic regions of *SLC23A1* and *SLC23A2* genes, but also by changes in the promoter regions, which have an impact on the binding of transcription factors, i.e., hepatic nuclear factor 1 (HNF-1) in the *SLC23A1* promoter [47,48] and miRNA sequences, i.e., posttranscriptional regulation of *SLC23A1* by miRNA in intestinal epithelial cells [49], as well as by epigenetic modifications of gene regulatory regions.

In addition to the SVCTs, vitamin C and its oxidized form dehydroascorbate (DHA) can be accumulated in the intestinal epithelium cells or transported to the bloodstream by glucose transporter (GLUT) as a result of facilitated diffusion across membranes. Particularly GLUT1, GLUT2, GLUT3, GLUT4, GLUT8, GLUT10 and GLUT14, the products of *SLC2A1* (MIM:138140), *SLC2A2* (MIM:138160), *SLC2A3* (MIM:138170), *SLC2A4* (MIM:138190), *SLC2A8* (MIM:605245), *SLC2A10* (MIM:606145) and *SLC2A14* (MIM:611039) genes respectively, have previously been described as DHA transporters [50,51,52,53,54,55]. All the aforementioned GLUTs are characterized by a highly homologous amino acid sequence, e.g., the GLUT2 protein sequence of the 524-amino acid has 55.5% identity with the GLUT1 product. Furthermore, the *SLC2A1*, *SLC2A2*, *SLC2A3*, *SLC2A4*, *SLC2A8*, *SLC2A10* and *SLC2A14* genes structure is very similar, which for the functional isoforms includes 10 exons (*SLC2A1*, *SLC2A3*, *SLC2A8*) and 11 exons (*SLC2A2*, *SLC2A4* and *SLC2A14*). An exception is *SLC2A10* gene which consists of five exons. GLUTs show tissue and cell-type specific expression. In particular, GLUT1, GLUT3 and GLUT10 have been shown to transport DHA, but it is *SLC2A2* and *SLC2A8* which are expressed in the small intestine cells on the highest level. In the literature, numerous variants of *SLC2A* genes have been described in terms of transport defect of glucose [54]. It is known that variants causing a lack of functional enzyme reduce the level of available DHA ex vivo [56]; however, up to date there has been no available data regarding the relationship between specific variants with altering circulating or tissue vitamin C level in human erythrocytes in vivo. It should be noted that in the course of investigation of substrates for the GLUT14 protein the Canadian scientists have identified three *loci* rs2889504, rs10846086 and rs12815313 in the *SLC2A14* gene associated independently with IBD [55]. These findings confirm the hypothesis concerning a genetically determined local dysregulation of dietary vitamin C or antioxidants transport which contribute to the development of IBD.

### 4.2. Genes Encoded Enzymes

In addition to transport, changes in the genes determining the antioxidant enzyme function and biotransformation constitute the other direction which should be taken into account in terms of genetic factors involved in the homeostasis of vitamin C. Metabolisms of vitamin C occurs in the liver and, to some extent, in the kidneys, in a series of reactions leading to the formation of the physiologically inactive 2,3-diketogulonic, xylonic or lyxonic acid excreted with urine [40]. Thus, in this group, such enzymes as such as NADPH-dependent oxidoreductases, glutathione S-transferases (GSTs), superoxide dismutase (SOD) should be mentioned. Research to date has been primarily investigating the relationship between vitamin C levels in the body and the polymorphism of the gene family of GSTs, which are able to reduce DHA to AA through enzymatic reactions with glutathione. Genetic information responsible for the synthesis of cytosolic GSTs is located on seven separate chromosomes. Each of eight classes identified in humans: alpha, beta, zeta, theta, mu, pi, sigma, and omega is most often coded by a group of genes [57]. So far, research of functional genetic variants of *GST*s impact on serum ascorbic acid deficiency has been referring to *GSTM1* (MIM:138350), *GSTP1* (MIM:134660) and *GSTT1* (MIM:600436) genes coding for GST enzymes in the mu, theta and pi classes, respectively. Dunsinska et al. were the first to demonstrate that the *GSTM1*-null genotype (homozygosity for a common gene deletion) was associated with increased circulating vitamin C levels and *GSTT1*-null genotype with reduced levels of vitamin C in the plasma [58]. Moreover, in this investigation, heterozygous individuals for the p.Ile105Val variant (c.313A>G, rs1695) in the *GSTP1* gene showed lower circulating vitamin C concentrations than the homozygous group. A Canadian researcher observed that individuals with GST null genotypes presented an increased risk of vitamin C deficiency if they did not meet the recommended dietary allowance which suggests that the GST enzymes protect against serum ascorbic acid deficiency when dietary vitamin C is insufficient [59]. However, this area certainly requires further extended research since the obtained associations have not always been fully consistent [60] due to the multiplicity of *GST*s genes. Currently, there is also no data regarding the impact of polymorphism of genes encoding other enzymes with antioxidant and biotransformation function on vitamin C levels in terms of human studies, which is an interesting direction and constitutes a challenge for future research.

## 5. Vitamin C and Bone Tissue in IBD Patients

Numerous nutritional factors play a role in the development of osteoporosis and the occurrence of bone fractures. Vitamin C is one of the nutrients which can affect BMD in various mechanisms [61]. On the other hand, nutritional deficiency of other vitamins and minerals, such as calcium and vitamin D, also influence bone mineral density (BMD) [38]. Moreover, the exclusion of other covariates is difficult and many research studies on the association between BMD and vitamin C intake have been observational. Nevertheless, vitamin C participation in proper bone mineralization is essential.

AA is vital for the synthesis and maturation of collagen, whereas vitamin C is a cofactor for prolyl and lysyl hydroxylases which perform a number of functions in the cell. These enzymes regulate the hydroxylation of proline and lysine in the collagen molecule which is essential for the synthesis and the maintenance of mature collagen [62,63]. Prolyl hydroxylase domain (PHD) protein, activated by vitamin C, may affect collagen structure and induce hydroxylates prolines of other protein, such as s hypoxia-inducible factors [23]. Furthermore, many studies, both on animals and on cell cultures, showed that vitamin C may participate in osteoclastogenesis and osteoblastogenesis [64]. In the study on cell cultures, AA caused increased levels of RANKL [65,66]. It is possible that vitamin C deficiency may stimulate the formation and differentiation of osteoclasts through the RANK/RANKL pathway. According to Le Nihouannen et al., AA increased numbers, size and nucleation of osteoclasts in primary mouse bone marrow cultures and monocytic RAW 264.7 cells in the early phase of osteoclastogenesis. Additionally, ascorbic acid accelerated of osteoclasts’ death in the late stage of osteoclastogenesis [67]. Therefore, vitamin C may have a twofold influence on osteoclastogenesis, i.e., it may both stimulate osteoclastogenesis and limit vitality of osteoclasts, which thus allows for a match of in vitro (stimulation of osteoclastogenesis) and in vivo (reducing bone resorption in proper provide of vitamin C) studies. The animal study showed that mice with a deficiency of vitamin C have a lower number and a decreased differentiation of osteoblasts [68]. According to Fraceschi et al. an incubation of mixture of osteoblastic-like cells with ascorbic acid stimulates the synthesis of a collagen-containing extracellular matrix and induces genes of protein related to the phenotype of osteoblasts, such as alkaline phosphatase (ALP), osteocalcin, osteopontin and osteonectin [62,69]. Additionally, vitamin C participates in the expression of a number of genes involved in the osteoblast activity, including growth, metabolism, communication and death. It is vital to notice that Vitamin C plays a role in the promotion of expression genes involved in the differentiation of chondrocytes throughout extracellular signal-regulated kinases (ERK). Moreover, AA induced activation of ERK in chondrogenic cells ATDC5, and an inhibition of ERK decreased chondrocyte differentiation induced by vitamin C [70,71]. Additionally, one of the factors of osteoporosis development is oxidative stress which may increase bone resorption throughout the activation of nuclear factor κB (NF κB) which is a crucial mediator of TNF-α and osteoclastogenesis [72,73]. In addition, vitamin C also may protect bone tissue from the influence on mediators of oxidative stress. AA reduces redox which removes singlet oxygen and reactive oxygen species [23]. However, no studies have reported the restorative effect de novo of vitamin C on the bone.nutrients-12-02263-t003_Table 3Table 3Genetic background for vitamin C deficiency in the human body.GeneSymbol, (MIM)Full NameVariantMAF in World Population *Impact on Vitamin C LevelStudy GroupReferencesRs Number (Minor Allele)Location*SLC23A1*(MIM:603790)solute carrier family 23 member 1rs10063949 (T)Intron42%Allele C association with an elevated circulating ascorbic acid in BWHHS, but an effect does not appear in a meta-analysis15 087 participants from 5 independent studies[45]No association with higher vitamin C plasma level300 subjects (150 POAG cases and 150 controls) from a Mediterranean population[74]AG and GG genotypes increase the risk of Crohn’s disease311 patients from the Manitoba IBD cohort[46]rs33972313 (T)Exon (p.V264M)4%Rare allele A associated with a reduction in circulating concentrations of ascorbic acid15 087 participants from 5 independent studies[45]G allele was associated with 11% higher plasma vitamin C97 203 white individuals including 10 123 subjects with ischemic heart disease[75]GG genotype was associated with a 9% higher plasma vitamin C compared with AA and AG combined106 147 individuals from the Copenhagen General Population Study[76]GA heterozygotes were associated with a 24% lower concentration365 cases and 1 284 controls from the EPIC cohort[77]rs11950646 (A)Intron38%GG or AG genotypes (compared with AA) were associated with a 13% lower plasma vitamin C concentration365 cases and 1 284 controls from the EPIC cohort[77]*SLC23A2*(MIM:603791)solute carrier family 23 member 2rs6133175 (G)Intron33%GG homozygotes associated with 24% higher plasma vitamin C concentrations365 cases and 1 284 controls from the EPIC cohort[77]C allele have a reduced risk of heart disease (implied that it is due to an increased vitamin C transport)97 203 white individuals including 10 123 subjects with ischemic heart disease[75]rs6053005 (T)Intron31%TT homozygotes associated with 24% higher plasma vitamin C concentrations365 cases and 1 284 controls from the EPIC cohort[77]rs1279683 (T)Intron45%GG subjects had a significantly lower plasma vitamin C concentrations than the other genotypes300 subjects (150 POAG cases and 150 controls) from a Mediterranean population[74]*GSTM1*(MIM:138350)glutathione S-transferase mu 1*GSTM1***0* (null)Whole gene deletion47.4% ^#^Higher vitamin C concentration in plasma115 individuals from Slovakia (44 survivors of myocardial infarction, 44 clinically normal controls and 67 population subjects)[58]4-fold increased risk of Vit. C deficiency for homozygote *GSTM1***0*/**0* and 2-fold for carriers of the *GSTM1**1 allele905 nonsmoking Canadian[59]A significantly lower level ofvitamin C in plasma compared with subjects carryingfunctional gene.388 volunteers from Slovakia[60]*GSTT1*(MIM:600436)glutathione S-transferase theta 1*GSTT1***0* (null)Whole gene deletion25% ^#^Lower vitamin C concentration in plasma115 individuals from Slovakia (44 survivors of myocardial infarction, 44 clinically normal controls and 67 population subjects)[58]12-fold increased risk of serum ascorbic acid deficiency for the *GSTT1*0/*0* genotype and only 2-fold for carriers of the *GSTT1*1* allele905 nonsmoking Canadian[59]Significantly lower level ofvitamin C in plasma as compared with subjects carryingfunctional gene388 volunteers from Slovakia[60]*GSTP1*(MIM:134660)glutathione S-transferase pi 1rs1695 (G)Exon (p.Ile105Val)35%Heterozygous individuals showed a significantly lower circulating vitamin C levels than homozygous GG115 individuals from Slovakia (44 survivors of myocardial infarction, 44 clinically normal controls and 67 population subjects)[58]No effect on serum ascorbic acid905 nonsmoking Canadian[59]No association with vitamin C in plasma388 volunteers from Slovakia[60]* based on 1000 Genomes project data (phase 3); ^#^ frequency calculated based on data of Kasthurinaidu et al. [78]; EPIC—European prospective investigation into cancer and nutrition; BWHHS—British women’s heart and health study; POAG—primary open-angle glaucoma.


## 6. Vitamin C in Various Diets

### 6.1. Mediterranean Diet

The Mediterranean diet (MD) is a dietary habit characteristic for the inhabitants of the Mediterranean region. A high intake of cereal products (couscous, pasta, bread), olives and olive oil, as well as grapes and wine is characteristic for this diet. Additionally, the consumption of fruits, vegetables, nuts, legumes, dairy products (yogurt, cheese), fish and meat (limited quantities) is in the usual range [79].

The study showed that the concentration of vitamin C increased when patients suffering from neoplasms have been on a Mediterranean diet. Moreover, the level of vitamin C was higher in the MD group than in the control group which is consistent with the guidelines of Cancer Society guidelines on nutrition and physical activity for cancer prevention [80]. A study by Hagfors L. et al. also indicated that individuals following the Mediterranean diet presented a higher amount of vitamin C than the control group. [81] It is vital to notice that MD patients consumed more fruits, although the amounts of vegetables and fruit juice were not significantly higher [82]. Thus, a higher consumption of vitamin C in subjects following MD is due to a high intake of vegetables and fruits which are the main sources of AA.

Diets including a high intake of sugar, sweetened beverages and a low intake of vegetables constitute a risk factor of UC [83]. It is possible that MD may protect from developing UC due to a high intake of vegetables. Tomasello G. et al. reported that the Mediterranean diet—rich in vegetables, fruits, olive oil and fish—may prevent from dysbiosis which is considered a risk factor for IBD [84]. Additionally, a high supply of vitamin C correlated negatively with the risk of CD development [85]. In fact, Marlow et al. reported that CRP decreased among patients with CD following six weeks of MD. Researchers noted the tendency (without significant changes) in growth and expression of *Bacteroidetes*, cluster IV and cluster XIVa and a decreased amount of cluster XIVa and Bacillaceae [86].

Adherence to the MD was associated positively with BMD of the femoral neck and lumbar spine [87]. Moreover, better MD-adherence was related with a lower risk of bone fracture [88]. Silva T. et al. reported that a better MD-adherence among women from the non-Mediterranean region also was associated with a higher BMD of the lumbar spine [89].

On the other hand, MD is considered to be a well-balanced diet and probably other nutritional factors may additionally affect bone tissue. The suggestion that proper intake of vitamin C is responsible for bone health and prevention IBD is virtually impossible. However, we think that MD may be recommended for patients suffering from IBD.

### 6.2. Vegetarian Diets

A vegetarian diet (VD) relies on the elimination of animal products, primarily meat. A variety of vegetarian diet is a vegan diet in which all animal products—including milk, eggs, honey—are eliminated. These diets have been associated with a higher risk of deficiency of certain nutrients, particularly protein, vitamin B12, iron, vitamin B2, iodine and n-3 fatty acid [90].

Individuals on the low-calorie, lacto–ovo–vegetarian diet consumed a higher amount of vitamin C than subjects following a standard low-calorie diet [91]. Among women on the vegetarian diet, the intake of vitamin C, folate and copper, was higher when compared to a non-vegetarian diet. On the other hand, vegetarians consumed a lower amount of vitamins B2 and B12, niacin, zinc and sodium [92]. Additionally, the serum level of vitamin C increased in individuals on VD [93].

Amarapurkar A. et al. reported that the VD constitutes a risk factor for CD and protects from UC [94]. The study does not show the impact of a vegetarian diet on the course of IBD. However, individuals on the vegetarian diet presented a lower psychological well-being [95]. VD may be favorable for patients suffering from IBD owing to the anti-inflammatory effect and the influence on rebuilding healthy gut microbiota [96]. Additionally, the maintenance of remission occurs more frequently in patients on a VD with a high-fiber content (32.4 g/2000 kcal) when compared with omnivorous individuals [97].

Furthermore, a vegetarian diet was a positive predictor of the total body bone mineral content and the total body areal bone mineral density. [98] Therefore, a decreased BMD of the femoral neck was higher in omnivorous subjects than vegans. However, the incidence of fracture did not differ between the groups. [99] The study showed that a low intake of protein food in terms of vegetarians was associated with a higher risk of wrist fractures [100]. Wang Y. et al. reported the lack of difference in BMD between vegetarians and non-vegetarians [101]. Moreover, the BMD of the femoral neck and lumbar spine was not different in vegans and individuals consuming meat [102]. On the other hand, a long-term study showed that vegan and vegetarian diets constituted risk factors for osteopenia in the femoral neck [103].

### 6.3. Low-Carbohydrates Diets

A low-carbohydrate diet (LCD) is based on a reduced carbohydrate supply to 50–150 g/day. A very low-carbohydrate diet containing carbohydrate below 20 g/day is called a ketogenic diet [104]. Simultaneously, a low-carbohydrate diet is high in protein and/or -fat.

Individuals following a low-carbohydrates diet (carbohydrates provided less than 45% total energy intake) consumed lower amounts of vitamin C, vegetables and fruits than the subjects in the control group [105]. Additionally, a diet based on guidelines of the Dukan diet (high protein diet, carbohydrates provide 10–25% of total energy intake) results in a lower vitamin C intake (2.5–4.5 mg) [106]. In fact, the ascorbic acid intake was 94 ± 59 mg in the Atkins diet. Nevertheless, it is vital to notice that the authors of the Atkins diet recommend a supplementation of vitamins and minerals. [107]

According to Reif et al. high cholesterol and fat intake, especially animal fat, was associated with a higher risk of UC development [18].

The incidence of CD was associated with an intake of fat, animal fat, n-6 fatty acids, animal protein, milk protein, as well as with the ratio of n-6 fatty acids and n-3 fatty acids [108]. Data regarding the association between n-3 fatty acids and CD are contradictory [85,109]. Jantchou et al. reported the association between the total protein intake and animal protein intake, as well as with the risk of IBD [110]. Moreover, there is a risk of CD associated with a meat intake [111].

An animal study has indicated that a 4-week high-fat diet with a standard and a low-protein supply decreased procollagen type 1 N-terminal propeptide (P1NP) when compared with the control group [112]. Kerstetter et al. have reported that a short-term (2 weeks) high-protein diet did not have an influence on bone balance [113]. Additionally, a low-carbohydrate diet may lead to acidosis resulting in an increased excretion of calcium without an increased absorption of calcium, since this element is possibly excreted from bone [114]. According to Draaisma et al. the BMD of children treated with the ketogenic diet and children in the control group was not significantly different [115]. On the other hand, Bergqvist et al. showed that bone mineral content (BMD) has decreased in children on the ketogenic diet [116].

### 6.4. Low-FODMAP Diet

A low-FODMAP (Fermentable Oligo-, Di- and Mono-saccharides and Polyols) diet assumes reduced the intake of products containing fermentable, poorly digestible oligosaccharide (fructans, galactooligosaccharides, fructooligosaccharides), disaccharides (lactose; only in patients suffering from lactose intolerance), monosaccharides (fructose) and polyols (sorbitol, mannitol, maltitol, xylitol, polydextrose, isomaltose). This further leads to the elimination of certain fruits (e.g., apples, nectarines, peaches, cherries, watermelons) and vegetables (onions, leeks, cauliflowers, asparagus). Low-FODMAP diet is used in the nonpharmacological treatment of irritable bowel syndrome (IBS) [117].

Individuals following a gluten-free and low-FODMAP diet consumed significantly higher amounts of vitamin C when compared with the individuals following a traditional gluten-free diet. [118] Moreover, there was no significant difference in vitamin C intake between patients on a low-FODMAP diet and subjects in the control group [119]. According to Eswaran et al. patients on low-FODMAP diet consumed higher amounts of vitamin C than patients in the control group, although the difference was not significant [120].

Pedersen et al. reported that symptoms of IBS decreased in patients suffering from IBD adhering to the low-FODMAP diet. [121] A meta-analysis has shown that the adherence to a low-FODMAP diet in patients with IBD decreased symptoms, such as abdominal pains, fatigue and nausea [122]. The low-FODMAP diet may be appropriate for patients suffering from IBD with functional gastrointestinal tract symptoms. However, patients should maintain a regular nutrients intake [123].

The study demonstrated that when compared with patients in the control group the patients on the low-FODMAP diet consumed similar amounts of calcium and vitamin D, which are essential for bone mineralization [124]. Another study showed that patients on the low-FODMAP diet consumed lower amounts of vitamin D and significantly lower amounts of calcium [120].

The comparison of various diets is presented in Table 4.

## 7. Gut Microbiota and Vitamin C in IBD Patients

Gut microbiota constitutes a subject of numerous studies, particularly in patients suffering from IBD. Quantitative and qualitative imbalances of microbiota are associated directly with the severity of the disease [125]. Clinicians made attempts to modify the gut microbiota composition for a clinical and endoscopic improvement [126]. The autoimmune process causing inflammation of the intestinal mucosa is the basis of IBD. One of the nutrients decreasing inflammation is AA. Vitamin C, as an antioxidant, creates a redox state and modulates gut microbiota. The study demonstrated that an increased vitamin C intake correlates with an increased *Firmicutes* and a decreased *Bacteroides* level [127]. Additionally, the supply of ascorbate decreases the abundance of *E. coli* and stimulates the growth of *Lactobacillus* and *Bifidobacterium* in the intestine. [128]. Vitamin C results in the restoration of the antioxidant state, favorable for gut microbiota and increases Coriobacteriaceae in pre-diabetes patients [129]. According to Pierre et al. changes in the body weight, mucosal levels of myeloperoxidase and syndecan-1 and levels of luminal IgA and IgG were observed in wild-type mice with deficiency of Il-10 following a diet supplemented with four nutrients (including vitamin C) [130]. In the clinical study, the association between the activity of Th17 in the pathway of Il-17 and the inflammation of the large intestine was revealed. As Chang et al. concluded, the ascorbic acid decreased the production of cytokines linked with Th17 [39]. According to the animal study by Yan et al. ascorbate decreased clinical symptoms, the level of the proinflammatory cytokine and the activity of myeloperoxidase (MPO) and malondialdehyde (MDA) in mice suffering from UC. Additionally, AA inhibited the expression of NF-κB, COX-2 and iNOS in the colon. The supply of vitamin C decreases oxidative stress and inflammation in the large intestine, which is particularly important for patients suffering from IBD [131]. The use of AA from goji berry in mice reduced colitis activity index, inhibited the activity of proinflammatory cytokines, as well as increased tight junction protein. Moreover, ascorbic acid modified gut microbiota, in particular Porphyromonadaceae, Prevotellaceae, Rikenellaceae, *Parasutterella*, *Parabacteroides* and which are the key bacteria related to IBD. In fact, this may be favorable for the course of the disease [132]. A high dose of vitamin C increases regulation of *MUC2* gene and simultaneously changes gut microbiota [133].

Table 5 presents certain bacteria which possibly affect BMD.

The numbers of fibroblasts are increased in inflamed mucosa of the intestine in the patients suffering from IBD, thus triggering the production of collagen type III. Vitamin C inhibits synthesis of collagen type III and stimulates the production of collagen type I which allows wound healing [134]. According to the animal study by Kondo et al. the use of vitamin C decreases levels of Il-6 and TNF-α. Nevertheless, the supplementation results are frequently contradictory, therefore, further research regarding the influence of vitamin C on the course of IBD is necessary [135].

## 8. Summary and Conclusions

Vitamin C has a pleiotropic effect on the human body. Ascorbic acid may have both a direct influence on cells and tissues, as well as an indirect impact by means of antioxidative mechanisms and the regulation of the gene expression. AA may affect bone tissue and play a role in the prevention of osteoporosis. Vitamin C is particularly important for patients suffering from IBD since AA may inhibit inflammation of the mucosa. However, the mechanisms of action of vitamin C remain unclear. Guidelines for the indications, dosage and the supplementation of ascorbic acid in certain diseases need to be clarified on the basis of the future experimental research studies.

## Figures and Tables

**Table 1 nutrients-12-02263-t001:** Concentration of vitamin C among inflammatory bowel disease (IBD) patients and healthy individuals.

Concentration of Vitamin C	Value
Ulcerative colitis [20]	39.1 ± 18.1 mmol/L
Crohn disease [21]	35.3 ± 25.8 mmol/L
Healthy individuals [20]	64.5 ± 12.5 mmol/L

**Table 2 nutrients-12-02263-t002:** Intake of vitamin C among patients suffering from IBD and heathy individuals.

Intake of Vitamin C	Value
Ulcerative colitis in active phase [20]	43.2 ± 27.6 mg/1000 kcal/day
Ulcerative colitis in remission [20]	67.5 ± 34.5 mg/1000 kcal/day
Crohn’s disease patients [22]	50.8 ± 40.3 mg/day
Healthy individuals [20]	49.5 ± 6.2 mg/1000 kcal/day

**Table 4 nutrients-12-02263-t004:** Comparison of various diets.

Diet	Vitamin C Content (Compared to Control)	Importance for IBD	Importance for Osteoporosis
Mediterranean diet	Higher	Protects	Prevents
Vegetarian diets	Higher	An increased risk of CD protects from UC	Contradictory data
Low-carbohydrates diets	Lower	Possibly an increased risk of IBD, because of a high intake of animal protein	Possibly an increased risk
Low-FODMAP diet	Higher	Appropriate for IBD patients with symptoms of IBS	Contradictory data

FODMAP—Fermentable Oligo-, Di- and Mono-saccharides and Polyols; CD—Crohn disease; UC—ulcerative colitis; IBS—irritable bowel syndrome.

**Table 5 nutrients-12-02263-t005:** Certain species that may prevent low bone mineral content (BMD).

Prevent Low BMD
*Lactobacillus reuteri* (animal study) [136]
*Lactobacillus rhamnosus* (animal study) [137]
*Lactobacillus helveticus* [138]
*Bifidobacterium longum* (animal study) [139]
*Bacillus subtilis* [140]

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
