# Peer review of "Vitamin C Deficiency and the Risk of Osteoporosis in Patients with an Inflammatory Bowel Disease"

_nutrients, 2020, doi:10.3390/nu12082263_

Round 1

Reviewer 1 Report

Introduction:

lines 45-46 change ‘so more’ to ‘Furthermore’,.. protects against mutations and prevents white blood cells from what? Is it oxidation or damage complete the sentence.

Lines 48-49 change sentence as “It is vital to note that food storage and processing decreases vitamin C content.”

Lines 49-52 – it is bit unclear what author wants to communicate. It is better to tell readers first what is the recommended daily requirement of vitamin C and clarify later the food portions which can deliver this amount of vitamin C!

Lines 52-53 replace ‘in the mechanism’ to ‘through facilitated diffusion..’

Lines 54 -55 please list common symptoms of scurvy such as bleeding and I think vision changes are rarely seen!

Vitamin C deficiency leads to scurvy which causes among other symptoms: fatigue, vision problems and neurological symptoms.

Line 55 Add ‘related to’ after are:

“Other symptoms are related to changes…”

Vitamin C deficiency in IBD and risk of osteoporosis

The authors are correlating low vitamin C levels with osteoporosis in IBD but this could simply be true for other vitamins as well! IBD is associated with risk of malabsorption and it is possible that it is the deficiency of  vitamin D/Calcium rather than vitamin C, which is associated with osteoporosis. There is no mention of adjusted analysis, if at all this is available, from different studies that the authors can justify such an association!

In addition, grammar needs improvement e.g. reactive oxygen species and not spices (line 72)

Vitamin C and bone tissue in IBD patients

It appears this relates to general association of vitamin C and bone health rather than a particular reference to IBD patients! It is unclear from this section whether the studies were in experimental animals or humans! The relationship between vitamin C and bone health is complex and is hard to prove in the absence of proper RCTs.

Vitamin C in various diets

It is hard to prove that increased vitamin C in Mediterranean diet prevents development of inflammatory bowel disease or leads to a higher BMD, in the absence of discussion suggesting whether other co-variates were taken into account in the quoted studies.

Reviewer 2 Report

1.- I want to suggest a little change in the title suppressing the initial A and writing simply "Vitamin C deficiency and risk of osteoporosis..."

2.- In the abstract and into the text you mus comment about the role of some drugs used in IBD like corticosteroids (Prednisone, Prednisolone and so on) and some biologics in the induction and maintenance of osteoporosis in these patients

3.- Inline 55, in the clinical description of the scurvy you must mention also the dermatologic problems because thew are very common and impressive

4.- In paragraph 2, please can you comment if the deficiency of Vit C is the same between CD and UC, and can you express the findings in a Table?

5.- The genetic background for Vit C deficiency can you summarize also in a Table?

6.-The influence of several diets is very interesting and their comparisons would be included in one Table for better comprehension to the readers

7.-Also for the influence of the microbiota would be better to include another Table signaling the bacterial type and their influence about the osteoporosis

8.- All the information collected from RTs, comparative studies, systematic reviews, and meta-analysis, would be included and remarked 

9.- The inclusion of the results of experimental studies may be useful 

10.- The conclusions are very clear

11.- The references are of good quality and recent

Reviewer 3 Report

nutrients-866861

The manuscript reviews an association between vitamin C (ascorbic acid) intake and bone marrow mineral density, and describes risk of vitamin C deficiency in the IBD patients, specifically Crohn’s disease and ulcerative colitis. They postulated the cause may be due to low intake of fresh vegetables and fruits and genetic factors responsible for vitamin C transport.

This is well-written review article. However, I would like to recommend to add figures/illustrates or graphic summary in each subtitle to explain an association between vitamin C (ascorbic acid) intake and bone marrow mineral density. Also, those summarizing the status of vitamin C-deficiency in the colon and bone of IBD patients are useful for readers to better understand.

If space is available, certain papers (Eur J Clin Exp Med 2017; 15 (4): 342–348; Digestion 2011;84:89–101; J. Nutr. Metab. Volume 2020, Article ID 2894169; and Redox Biology 6: 617-639, 2015) can be cited.

Round 2

Reviewer 1 Report

Abstract

Line 27 change inflammatory bowel diseases to inflammatory bowel disease and add ‘a’ after at

Line 29 change ‘due to such as factors as reducing’ to ‘due to factors such as reduced’

In general inflammatory bowel diseases should be changed to inflammatory bowel disease throughout the text including figure!

Introduction

Line 42 delete ‘of’

Line 44 change sentence to ‘Ascorbic acid acts as an electron donor for both enzyme and non-enzyme reactions’

Line 46 add ‘against’ before mutation

Line 50 change ‘decrease’ to decreases’

Line 55 ‘and constitutes about …’ change to ‘and only 33% of vitamin C is absorbed when dose is increased to 1250mg’.

Line 59 add ‘and’ before neurological disorders.

Lines 59-60 Change ‘However, most commonly the deficiency’ to ‘Manifestations of vitamin C deficiency are usually related to…’

Line 65 change ‘and reducing the’ to ‘and reduction in consumption’ and ‘fruit’ to ‘fruits’

Line 65-66 – change sentence to ‘The serum levels of vitamin C in patients with inflammatory bowel disease and healthy individuals are presented in Table 1.’ In addition, are these mean or median vitamin C levels?

Change sentence in lines 67-68 as ‘Patients with IBD have high TNFα levels, which also decreases absorption of various nutrients, including vitamin C.’

From Table 2 it appears healthy individuals have almost similar intake of vitamin C when compared to patients with IBD. Minor differences could be by chance and it would be appropriate to give statistical significance if available otherwise it is a mere speculation!

Line 86 correct ‘spices’ to ‘species’

Authors commented that a meta-analysis showed that higher vitamin C was associated with a higher BMD – what was the effect size and needs numbers with confidence intervals! In addition the authors comment that women suffering from osteoporosis consumed lower amount of vitamin C than those with normal BMD, however, they also had vitamin D deficiency! These statements do not prove that with confidence that vitamin C is linked to a higher BMD and needs clear justification.

Lines 96 change ‘although AA has not high toxicity to ‘Although, AA has minimal side effects and is well tolerated, high doses…..’

Line 97 a total daily dose of what number is it 1000mg or more should be included so that readers know what doses increases risks of renal stones and change ‘stone’ to ‘stones’

Lines 104 add ‘agents’ after biological

Lines 112 change ‘decrease calcium absorption’ to ‘decreasing calcium absorption’ and ‘inhibit’ to ‘inhibits’

Line 115 correct ‘anty-TNF’ to ‘anti-TNF’

Lines 116 -117 are unclear about the role of anti-TNF!

Lines 118 change ‘a decreased activity’ to ‘a reduction in activity’

Line 125 change ‘show’ to ‘increase’
